# Investigation of Multiparameter Laser Stripping of AlTiN and DLC C Coatings

**DOI:** 10.3390/ma14040951

**Published:** 2021-02-17

**Authors:** Tomáš Primus, Josef Hlavinka, Pavel Zeman, Jan Brajer, Martin Šorm, Adam Čermák, Pavel Kožmín, František Holešovský

**Affiliations:** 1Department of Machining, Process Planning and Metrology, Faculty of Mechanical Engineering, Czech Technical University, 166 07 Prague, Czech Republic; holesovf@kto.zcu.cz; 2Department of Production Machines and Equipment, Faculty of Mechanical Engineering, Czech Technical University, 128 00 Prague, Czech Republic; J.Hlavinka@rcmt.cvut.cz (J.H.); P.Zeman@rcmt.cvut.cz (P.Z.); J.Brajer@rcmt.cvut.cz (J.B.); 3Hofmeister s.r.o., 301 00 Plzeň, Czech Republic; sorm@hofmeister.cz (M.Š.); cermak@hofmeister.cz (A.Č.); kozmin@hofmeister.cz (P.K.)

**Keywords:** laser stripping, AlTiN, DLC C, coating, nanosecond laser

## Abstract

The lifetime and properties of cutting tools and forming moulds can be prolonged and enhanced by the deposition of hard, thin coatings. After a certain period of usage, the coating will deteriorate. Any remaining coating must be removed prior to successful recoating. Laser stripping is a fast and environmentally friendly coating removal method. In this paper, we present laser removal of two types of coatings deposited on a 1.2379 tool steel substrate, namely, an AlTiN coating with high hardness and a DLC C coating with a small coefficient of friction (COF). A powerful nanosecond laser was employed to remove the coating from the substrate with high efficiency, along with suitable residual surface roughness. Measurements were taken of surface roughness, removed depth, and working time on a stripped area of 1 cm^2^. The samples were evaluated under a microscope, with a 3D profilometer, and by EDS chemical analysis. Successful removal of the coating was confirmed by optical analysis, but detailed chemical characterisation showed that about 30% of the coating element may remain on the surface. Moreover, a working time of less than 7.5 s per cm^2^ was obtained in this study. In addition, it was shown that the application of a second low energy, high frequency laser beam pass leads to remelting of the peaks of the material and reduced surface roughness.

## 1. Introduction

To increase manufacturing efficiency, cutting tool and mould surfaces are covered with different types of thin, hard coatings [1]. After tools incur damage from cutting processes, they can be sharpened and reused, but no coating residue should remain. Proper coating removal is important for successful sharpening, redeposition of a new coating, and the final properties of the reground and recoated tool [2]. Regrinding cutting tools has the potential to save up to 70% of the cost of new tools (depending on the parameters used, such as cutting speed), but their durability is slightly lower than that of new tools [3].

Coatings are used as a protective barrier against wear, friction, abrasion, adhesion and thermal damage [4]. Some coatings can also decrease friction, therefore increasing tool lifetime [5]. On the one hand, coatings frequently include TiN, which increases wear resistance. On the other hand, coatings consisting of AlTiN have a high hardness. CrN is another type of coating, which increases corrosion resistance and provides lubricity [2]. Coatings made of synthetic diamonds (diamond-like coatings, DLC) with low friction and/or a high hardness are rapidly evolving [6]. In addition, the use of DLC coatings for biomedical applications has been discussed recently [7].

Hard coatings, such as TiN, CrN and DLC, are used in injection moulds for similar reasons. Moulds are commonly made of tool steel. Coatings shield the mould from abrasion and adhesion, thereby prolonging its life [8]. After regrinding, a new coating is deposited. Residual coating may increase or decrease the adhesion of the new coating and also affect the quality of the regrinding [9].

Coatings are most commonly removed by chemical methods or mechanically. Chemically (or electrochemically) stripped substrates may have a roughness similar to that of the coated material [10,11]. The disadvantages of chemical methods include long decoating times and large volumes of waste, which may be toxic [12]. These problems were partially resolved by the introduction of electrochemical dissolution, which is both faster and safer for the environment [10,13]. However, issues with waste disposal and substrate material damage remain. Moreover, the widespread use of AlTiN coatings [14] increases the need for rapid and high-quality coating removal. At the same time, the application of DLC coatings is increasing. DLC coatings can be removed using conventional methods, but with a high risk of substrate damage. As a result, laser stripping has become the leading DLC coating removal method. Laser stripping is a more ecologically friendly, effective method, and is associated with minimal damage to the substrate [15].

Marimuthu et al. [16] reported using a KrF excimer laser with a wavelength of 248 nm (UV light) to remove 2 µm of CrTiAlN coating from a steel substrate. The decoated area had a surface roughness (Ra) of approximately 0.415 µm with minimal damage to the substrate. Marimuthu et al. [17] used the same KrF laser to remove 2 µm of TiN coating from a tungsten carbide substrate. Online monitoring of the stripping process was used to scan the coating removal and control the quality of the process. Moreover, the ablation threshold of the TiN coating was determined [17]. Marimuthu et al. continued these experiments on a similar basis to study TiAlN coatings [18]. A similar TiN coating was stripped using a different method in a study presented by Hu Ch. et al. [19], where a TiN coating was deposited on a Ti6Al4V alloy and stripped using thermal and force effects of laser shock. However, this approach led to significant melting, cracks, and pit formation, which are undesirable effects for industrial processes. Long See et al. [20] studied the effect of combining different fluences and number of UV laser pulses on TiAlN coatings. They also studied the effect of the laser beam on the original substrate and the presence of micro cracks on the substrate surface. Ragusich et al. [21] studied decoating of TiAlN on aerospace components with a thickness of 20 µm. They compared the use of an excimer laser and a solid state Ti:sapphire laser for removal of TiAlN coatings. Zivelonghi et al. [22] presented successful removal of a DLC coating by laser stripping; several low-power passes were used to gradually remove a CrC/Cr–Cr:DLC–DLC coating with a thickness of 6 µm. Additionally, Assurin et al. [23] presented KrF excimer laser removal of a multilayer coating consisting of 4 μm TiCN + Al_2_O_3_ and 3.2 μm CrN + DLC layers. In this case, removal of only the outer CrN + DLC layer is presented.

Although coatings were successfully removed in the aforementioned papers, none of them reported the time required for coating removal. The aim of this paper is to present a method of laser stripping for highly productive removal of hard coatings, namely, AlTiN and DLC. The main goals of the experiment were to achieve a short processing time, good quality and suitable surface roughness for any new coating, as well as to provide a detailed analysis of coating residues and laser radiation parameter effects on the quality of the stripped surface. In addition, a nanosecond Nd:YAG laser was used in this experiment, which is applicable to industry.

## 2. Materials and Methods

Two types of coatings, DLC C and AlTiN, were chosen for this experiment. AlTiN is used for a variety of cutting tools and moulds. AlTiN coatings have a hardness of approximately 3300 ± 300 HV and are used for universal abrasion protection. DLC C coatings have a hardness of 900 ± 50 HV, but have a very low coefficient of friction (COF) against steel (0.08). They are used for sliding connections inside machines and the reduction of friction in cutting tools [24] and moulds [25], and to increase corrosion resistance [26]. DLC C coatings are multi-layered, consisting of multiple a-C:H layers at the top with a total thickness of 1.1 µm, and are combined with a chrome-based CrN layer with a thickness of 1.5 µm, making a total coating thickness of 2.6 μm. The second C in the name of the coating is redundant and only indicates that there are no impurities in the coating. The addition of metal atoms such as Ti, W, Crz, Zr, Cu or Ag to the carbon film can change some properties of the coating, such as a reduction in residual stress or better adhesion of the coating to the substrate [27]. The thickness of the coatings and their adhesion to the substrates were evaluated by the Calotest and scratch test. Other properties of the coatings were supplemented according to the data sheets provided by the coating manufacturer. The characteristics of both coatings are listed in Table 1.

The substrate steel chemical composition (from the data sheet) is shown in Table 2.

The coatings were deposited on a disc specimen with a diameter of 25 mm and a thickness of 4.4 mm. The discs were mirror polished with a diamond paste containing 3 µm grains, yielding a final surface roughness Ra = 0.005 ± 0.001 and Rz = 0.02 ± 0.005.

A solid state Nd:YAG pulsed laser was used to strip the coatings. The laser has a wavelength of 1064 nm and a pulse duration of 120 nanoseconds with a repetition rate ranging from 1 to 50 kHz and a spot diameter of 0.18 mm. It contains a galvo scanner with a maximum scanning speed of 3000 mm·s^−1^. It uses a non-polarised Gaussian beam. This laser device is suitable for cleaning and engraving. The software that was used is capable of measuring working time. The specifications of the laser device are listed in Table 3.

### 2.1. Experiment Specifications

In the experiment, the laser beam parameters were changed, that is, the repetition rate, average power and scanning speed. Thus, the peak energy (Ep), peak power density (P_0_) and fluence (F) varied. The peak power density used to remove an AlTiN coating presented in a paper by Long et al. [20] was 1.49 × 10^8^ W·cm^−2^ (for 10 passes) with a fluence of 0.68–7.44 J·cm^−2^ on the same type of coating. However, due to the much longer pulse duration of the laser source (120 ns) and different spot size (0.18 mm), the fluence needed to be increased. The fluence was increased to a higher range of 2.9 to 29 J·cm^−2^ by increasing the power and lowering the frequency to below 10 kHz. The goal was to remove the coating in one pass and decrease the working time per cm^2^, considering that higher power density leads to faster coating removal, in accordance with [24]. However, as the faster scanning speed may cause a decrease in surface quality, the experiment addressed the trade-off between speed and quality (the decrease of which is directly related to pulse duration and the heat affected zone (HAZ)) [28]. The parameters used to strip the AlTiN coatings are shown in Table 4.

The idea of using a second low energy pass to increase the surface quality was tested for sample AlTiN4. It was assumed that a second pass with low energy would remove a small amount of material and partially melt the peaks of material in the stripped area, thereby improving the surface roughness. The melted material then solidifies again into a form with a smoother surface, thus reducing the surface roughness. Indeed, the same technique is used for laser polishing [29].

The parameters used for sample AlTiN2 were also used for sample DLC3, but were varied for all other samples. Sample DLC4 was treated with a second high frequency pass. The rest of the parameters used to strip the DLC-C coatings are shown in Table 5.

For all of the samples, the decoating area was set to 5 mm^2^ (2 mm wide and 2.5 mm long) and the laser beam moved line by line in the defined overlap. The working time was measured using WMARK laser operating software (WMark 1.1; MediCom, a.s.; Prague; Czech Republic). The measured time is the operating time of the beam in the laser stripping process. To minimise the Gaussian beam effect, proper spot overlap (Figure 1) in both scanning directions is needed. For this experiment, a constant spot overlap of 70% was chosen in both directions based on previous experience. Preliminary laser stripping tests on the same samples resulted in surface destruction when using a higher percentage overlap, due to a higher HAZ. Moreover, a spot overlap of 70% allows the use of a higher scanning speed in comparison with higher overlaps [28]. In the parallel direction (the direction of the laser beam motion), the overlap (H_x_) is defined as a function of scanning speed (v), frequency (f) and beam diameter (D). The beam diameter in this experiment was 0.18 mm. The traverse (side) overlap (H_y_) is set in the software. For a 70% spot overlap, the shift (S_*x*_) between pulses had to be 30% of the diameter—in this case, 0.054 mm. The overlap equations are as follows:(1)$$H_{x}=\left(1-\frac{v}{D\cdot f}\;\right)\cdot 100\;\left[\%\right]$$
(2)$$S_{x}=D\cdot \left(1-\frac{H_{x}}{100}\right)\;\;\left[\mathrm{mm}\right]$$

### 2.2. Surface Analysis

The laser-stripped samples were analysed using a HEIScope optical microscope (Howard Electronics, El Dorado, KS, USA) with a Navitar objective and the final surface after stripping was compared to an image of untreated coating (Figure 2). The stripping was considered a success when the reflection of light from the steel was visible over the entire stripped area, similarly to the uncoated surface.

A second analysis was performed with a Zygo NewView 7200 3D relief meter (Lambda Photometrics Ltd, Luton, UK), which was used to measure the surface roughness and the maximum depth of the removed layer. A Zeiss field emission scanning electron microscope (FESEM; ULTRA PLUS, Oberkochen, Germany) equipped with an energy-dispersive spectrometer from Oxford Instruments (EDS; X-Max 50, Oberkochen, Germany) was used for the third analysis. EDS/SEM provided chemical analysis of the surface. The weight percentage of each element in the surface layer was obtained from a square area with a side length of 200 µm.

## 3. Results

### 3.1. Laser Stripping of AlTiN Coating

#### 3.1.1. Surface Morphology

The stripping depth results and the working times for each parameter are shown in Figure 3. The shortest stripping time was 5.34 s for an area of 1 cm^2^, which was achieved using the parameters for sample AlTiN2. The longest working times were 7.7 s for the parameters of sample AlTiN1 and 10.1 s for AlTiN4, where two laser beam passes were applied. In all cases, the maximum depth of the removed layer exceeded the thickness of the AlTiN coating (3.2 µm). Sample AlTiN2 had the smallest removed depth (4 μm), while the biggest removed depth (5 μm) was measured for sample AlTiN4. Both AlTiN3 and AlTiN4 were produced with the same first pass energy, but the second pass used for AlTiN4 led to an increase in depth of about 0.6 µm. In summary, an increase in fluence does not lead to a constant increase in the removed depth. Despite the fact that the largest removal was for the largest fluence, for the smallest fluence the removal was the second largest. Moreover, higher fluence affected the surface morphology due to greater surface melting.

The surface roughness measurement results are shown in Figure 4. Generally, the roughness in the X axis (the direction of the laser beam motion) was lower than the roughness in the Y axis. A ridge of material formed in the crossover between two lines of pulses, increasing the roughness in the Y axis. This is because of a significant heat affected zone. The surface roughness seems to be at a minimum for sample AlTiN2, achieving coating thickness with the second lowest processing time. Taking into account all mean values of the measured data and their standard deviations (presented by error bars), the surface roughness varied with great uncertainty. This may be caused by varying HAZ, or non-homogeneity of the substrate. Uncertainty seems to be slightly reduced after the second pass.

The 3D characteristics of the surface are shown in Figure 5, where the peaks of the melted material can be seen. It is assumed that the peaks were formed by coating remnants. In the image of AlTiN1 (Figure 5a), the material has the largest volume of ridges, corresponding with the fact that sample AlTiN1 had the highest roughness value. In AlTiN2 (Figure 5b), there are small peaks and significantly larger valleys—cleaned spaces—in comparison with sample AlTiN1, where the peaks are significantly higher. This demonstrates the lowest surface roughness. There are a high number of peaks in sample AlTiN3 (Figure 5c), however, they are sharp and thin. In the image of AlTiN4 (Figure 5d), the peaks are rounded down, and the valleys between ridges are deepened. This was caused by a second high frequency pass that melted the peaks and ablated the cleaned spaces. Consequently, this caused an increase in Ra-Y. All of the images may be affected by parasitic light reflection from CSI (Coherence scanning interferometry). The samples were further analysed with EDS/SEM.

The surface roughness parameters Sa, Sz, Ssk and Sku are presented in Table 6. The results of the area roughness correspond to the results of the plane roughness. The lowest surface roughness was measured for sample AlTiN2. The Ssk (Skewness) represents the degree of bias of the roughness shape. In our case, the Ssk values are higher than zero, which means that the height distribution is skewed below the mean plane. The Sku (kurtosis) value represents the sharpness of the roughness profile. In our case, the Sku values are higher than three for AlTiN2 and AlTiN4, which means that the height distribution is spiked. For AlTiN1, the height distribution is skewed above the mean plane, and for AlTiN3, the height distribution can be marked as normal [30].

#### 3.1.2. Chemical Characterisation

The results of the EDS/SEM analysis demonstrate the spectrum of elements and their weight percentage, along with the distribution of chemical elements on the surface. A detailed view of the stripped surface can be seen in the SEM images presented in Figure 6. These images show that there are no cracks on the surface; only high melting and solidification can be seen in all images. Figure 6d shows the melting of the surface caused by the second pass.

The presence of substrate steel elements, namely, iron, carbon, manganese, molybdenum, silicon, chromium and vanadium was an expected result. At the same time, it was expected that the weight percentage of the coating elements—aluminium, titanium and nitrogen—would be close to zero. The amount of carbon contamination on the surface is not included in the analysis results.

The results of the analysis show that there are coating residues on the stripped surface. Due to the residual oxide generated by melting and other impurities on the decoated surfaces, a larger amount of residual coating was measured.

The detailed composition of chemical elements on the surface is described in Table 7. The weight percentage of AlTiN coating on the surface layer indicated coating residues of about 20–30%.

The lowest amount of coating elements was found for sample AlTiN2, the same sample with the lowest surface roughness. In comparison with the optical image of the stripped AlTiN2 surface, where the substrate steel can be seen on the whole surface, the EDS demonstrated there was a high amount of coating residue. This may be caused by the small area of EDS measurement, where the focus was likely more on poorly stripped areas. These results indicate that an optically stripped surface can still contain a large amount of coating elements. It follows that the validation of the laser stripping process should concentrate mainly on optical analysis accompanied by chemical analysis.

The analysis of samples AlTiN1 through AlTiN4 showed that the percentage of vanadium on the surface is very small, and it was not included in the results. Moreover, the weight percentage of residual coatings increased as the fluence increased, contrary to the first assumption that a higher fluence leads to more effective coating removal. Sample AlTiN4, which was treated by a second high frequency pass, had a larger weight percentage of Ti and Fe than sample AlTiN3 fabricated by the same energy in the first pass.

However, all of the surfaces contained a significant percentage of oxygen, in the form of oxides with the elements mentioned above. The oxides are most likely waste generated during the laser stripping process. On the other hand, all of the tested samples were stripped with a processing time of a few seconds, so, the nanosecond laser stripping process is very fast and efficient.

In summary, by comparing the sums of the substrate and coating elements, approximately 70% of the AlTiN coating was removed by the laser stripping process.

### 3.2. Laser Stripping of DLC C Coating

#### 3.2.1. Surface Morphology

The DLC C coating laser stripping experiment was set up identically to the AlTiN coating stripping experiment, and the results were evaluated using the same methods. The measured time and removed depth of the DLC C coatings are shown in Figure 7.

The shortest coating removal time for 1 cm^2^ was 5.4 s for samples DLC1, DLC2 and DLC3. The longest time needed to remove 1 cm^2^ was 7.8 s, which included two passes. In all cases, the maximum depth of the removed layer exceeded the thickness of the DLC C coating (2.7 µm). The smallest removed depth of 5 μm was measured for sample DLC1, which was fabricated with the smallest fluence. In contrast, sample DLC4, produced with two passes and the highest energy, resulted in a maximum removed depth of 5.6 μm. The depth increase was only 0.6 µm although twice as much energy was used. Examining the influence of the second high frequency pass, it was observed that there was a small increase in removed depth for quite a large increase in working time. Figure 7 also makes it apparent that the removed depth does not increase as the fluence increases and stays constant in comparison with the AlTiN coating removal. This may be due to the different ablation behaviour of the DLC C multilayer coating.

Figure 8 presents a comparison of the surface roughness measured in the X and Y directions. The error bars represent the standard deviation of the mean taken from 10 measurements in the same direction. The worst surface roughness was measured for sample DLC3 with the same Ra in both measured directions. The best surface roughness was observed in sample DLC4 along with sample DLC1. A comparison of samples DLC3 and DLC4 shows the effect of the second pass on the remelting of the peaks of the molten material, leading to improved surface roughness. Moreover, including a second pass also correlates with reduced uncertainty of the measurements.

Four images of the stripped areas were obtained using the Zygo equipment, and are presented in Figure 9. There is no visible pattern for sample DLC1 and the peaks are scattered and high. Sample DLC2 shows a pattern and the peaks are rounded. Sample DLC3 exhibits the highest roughness, which is demonstrated in the image by high peaks. In the image of sample DLC4, the peaks are smaller than in the DLC3 image, demonstrating that sample DLC4 has the best surface roughness value and also showing the benefit of a second pass.

The surface roughness parameters Sa, Sz, Ssk and Sku are presented in Table 8. The lowest surface area roughness was measured for sample DLC4. The Ssk values are higher than zero for all samples, which means that the height distribution is skewed below the mean plane. The Sku values are higher than three for all samples except DLC2, which means that the height distribution is spiked. The height distribution is normal for DLC2 [30].

#### 3.2.2. Chemical Characterisation

The detailed chemical composition of the samples after laser stripping of the DLC C coatings is shown in Table 9. A detailed view of the stripped surface can be seen in the SEM images shown in Figure 10. There are no cracks in the surfaces except for DLC4, where crack formation can be observed. The cracks are about 200 nm wide and were probably formed during the second low-energy pass, given that no cracks are observable in DLC3, which was created using only one laser beam pass.

From the EDS analysis, it is apparent that the DLC C coating was probably not fully removed. The amount of carbon was higher than 15%, the amount of chromium varied from 12 to 40%, and the percentage of nitrogen was around 0.7% for all of the measured samples. This leads to the conclusion that the multilayer coating may have remained on the stripped samples. However, it is important to note that chromium is also contained in the substrate steel and that the presence of carbon can be caused by air pollution. There was also a significant percentage of oxygen, ranging from 8 to 14%. Moreover, the weight percentage of the remaining coating increased with increasing fluence.

It appears that the top layer of the coating was removed and the rest of the carbon was from other coating layers or from the ambient air. In the analysis results for sample DLC4, which was fabricated by two laser passes, it is evident that no nitrogen is present. This may mean that the layers of a-C:H and CrN were successfully removed, but there is still a significant amount of chromium residue on the surface, probably in the form of chromium dioxide. In this case, a second high frequency pass increased the oxidisation of the surface.

## 4. Discussion

In this paper, laser stripping of two different hard coatings, namely, AlTiN and DLC C, was presented. The goal was to completely remove the coating from the substrate steel in a highly productive manner with one laser beam pass, resulting in good surface quality.

The working laser time for the decoating of AlTiN and DLC C coatings was lower than 7.5 s to strip 1 cm^2^. In comparison, Zivelonghi et. al. [22] used 10 laser beam passes to strip a DLC coating with a scanning speed of 200 mm·s^−1^. In another study, Marimuthu. et al. [16] used a scanning speed of 4.2 mm·s^−1^, which is 100 times lower than the speed used in this paper.

The lowest surface roughness was achieved for sample AlTiN2. For the DLC C coating, the best surface roughness was achieved for sample DLC4. For this sample, a second low energy pass was used to decrease the surface roughness. Compared to the same sample without a second pass, the roughness Ra was reduced by 0.1 µm in the X direction and by 0.2 µm in the Y direction. The same idea was not confirmed for the AlTiN coating, where the surface roughness Ra increased when using a second pass. The difference in roughness in both directions is caused by the ablation method. It can be reduced with different scanning strategies, but applying more than one laser pass. The resultant roughness Ra of 0.4–0.8 μm seems to be sufficient and would thus be a suitable basis for further studies investigating the influence of substrate roughness on redeposition of a new coating. The best achieved roughness Ra = 0.4 ± 0.08 μm is similar to the surface roughness Ra = 0.415 achieved by Marimuthu et al. [16] after stripping TiAlCrN/AlTiN coatings.

The roughness measurements on all surfaces showed great uncertainty. However, the formation of a significant HAZ is to be expected with the use of a Nd:YAG laser [31]. As previously reported for a similar solid-state, nanosecond laser, the HAZ can reach a size of up to tens of micrometers [32]. This will supposedly cause a decrease in surface quality. The idea of additional surface treatment was formed in anticipation of a change in quality. Similar to laser polishing [29], a second, low-energy pass was used to increase surface quality. This phenomenon was effectively demonstrated with the DLC coating, but not so much with the AlTiN coating. The second pass of a laser beam can also cause some changes in surface chemistry or tensile stresses of the underlying steel, especially after melting and solidification, as described. According to [15], there were no measured tensile stresses in the surface layers after laser irradiation (with significant melting and solidification).

Despite the fact that the measured depth of all of the samples was higher than the thickness of the coating, coating residues remained on all of the surfaces, as confirmed by EDS analysis. This is probably due to the significant melting caused by the use of a nanosecond laser, or possible redeposition of the ablated substrate back into the stripped area. Due to a significant HAZ, the substrate is also affected. Residual thermal stress can cause cracks in the substrate; however, crack formation was observed only in the DLC4 surface. In this case, the substrate was affected by the high energy of the second laser pass. Crack formation was also observed by Long See et al. [20]; however, they used WC-Co as a substrate, so ablation of Co may have resulted in crack formation in this instance.

The suitability of optical or depth measurement for evaluation of a correctly stripped sample should be discussed. Using optical analysis, the coatings appear to be stripped because of the reflections of the underlying steel. However, as our research has shown, it is essential to study the surface chemistry because there are still coating residues on the apparently completely stripped surface.

The residues of coating were evaluated by EDS and approximately 22–32% of AlTiN and approximately 20–30% of DLC coatings remained on the stripped surface. However, the analysis of the residues of both coatings were not entirely accurate due to the presence of chromium and carbon in the coating, as well as in the underlying steel. In addition, it was proved by Marimuthu et al. [18] that re-deposition of an ablated coating can be affected by the composition of the stripped area. Although there were coating residues on the surface, according to the coating provider, no problems were observed with applying a new, identical coating to the stripped samples.

Moreover, based on assessment of the fluence dependence, it is apparent that the DLC coating needs lower energy for ablation than the AlTiN coating. This may be due to the better toughness and higher thermal stability of the AlTiN coating. Compared to previous studies [16,17,18,20], the scanning speed employed here is significantly faster, leading to a shorter processing time with a reduction in coating removal costs.

A relationship between depth and fluence was not observed in the laser stripping of AlTiN, but in the DLC C coating, despite the large uncertainty, an increase in laser fluence led to a constant increase in removal depth. This may be due to the different properties of each layer in the DLC C multilayer coating. The non-linearity of the AlTiN stripping process can be caused by melting at high fluences.

The analysis of area surface roughness highlights the necessity of changing the scanning strategy, for example, by changing the spot overlap, adopting a hatching strategy, or using more laser passes with lower energy.

## 5. Conclusions

In summary, in this paper we have presented a method of laser stripping for rapid decoating of AlTiN and DLC coatings with good surface quality for the recoating process. We achieved very short working times with the successful coating removal evaluated by optical analysis, with one or two passes of the laser beam. According to EDS, approximately 20–30% of the coating elements were detected on the stripped surfaces. Despite this fact, this study also demonstrated that it is essential to study surface chemistry along with the optical appearance of the surface in order to evaluate a properly removed coating.

## Figures and Tables

**Figure 1 materials-14-00951-f001:**
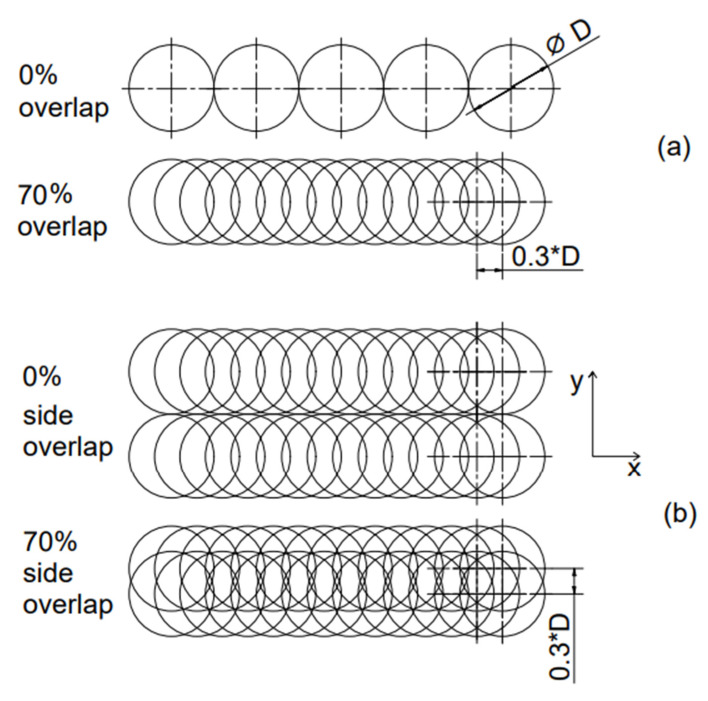
Scheme of a pulse overlap, (**a**) overlap in the parallel direction—scanning direction, (**b**) overlap in the side direction.

**Figure 2 materials-14-00951-f002:**
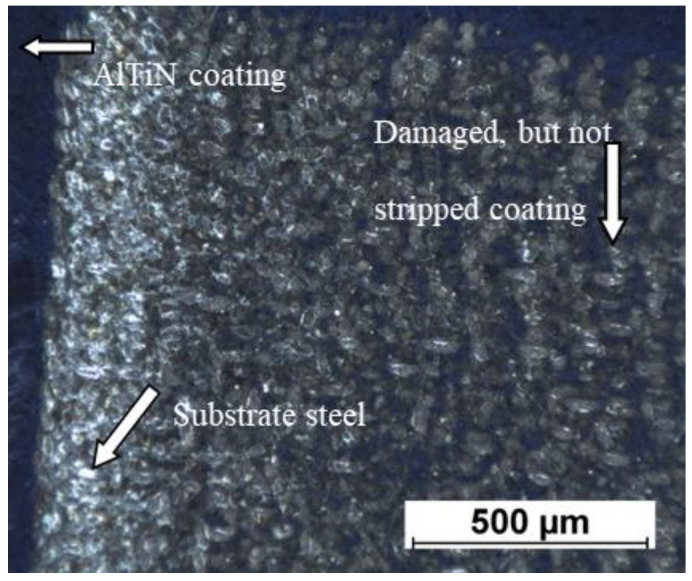
An example of an optical image of a laser-stripped sample.

**Figure 3 materials-14-00951-f003:**
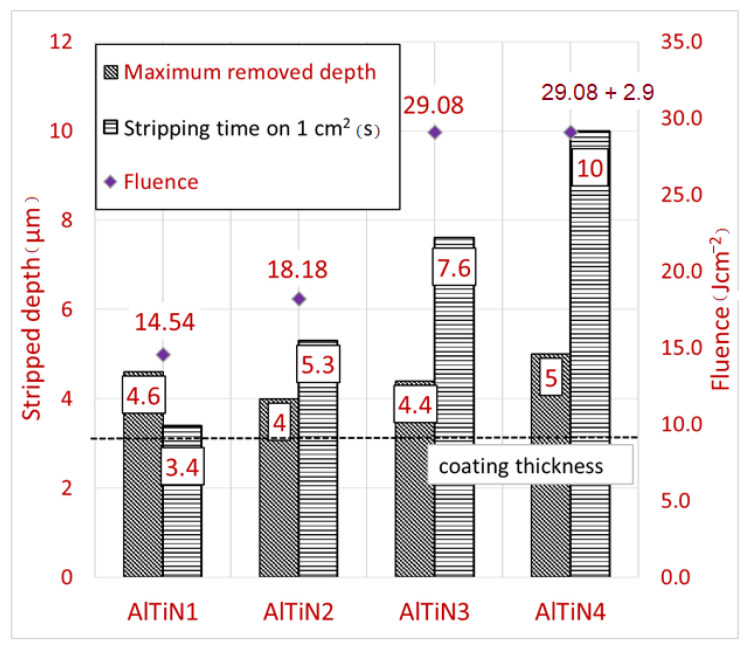
Bar chart of removed depth and working time for AlTiN coating stripping.

**Figure 4 materials-14-00951-f004:**
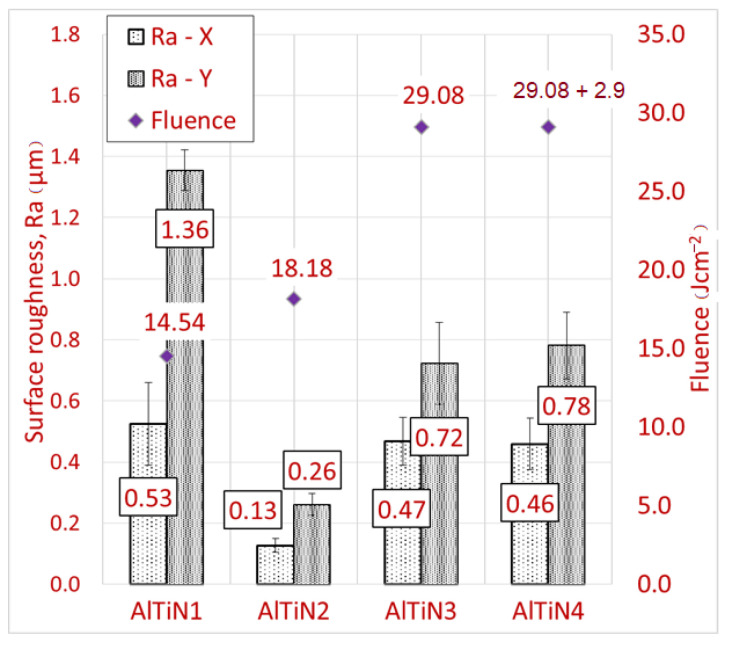
Bar chart of surface roughness after laser stripping of the AlTiN coating in relation to the fluence.

**Figure 5 materials-14-00951-f005:**
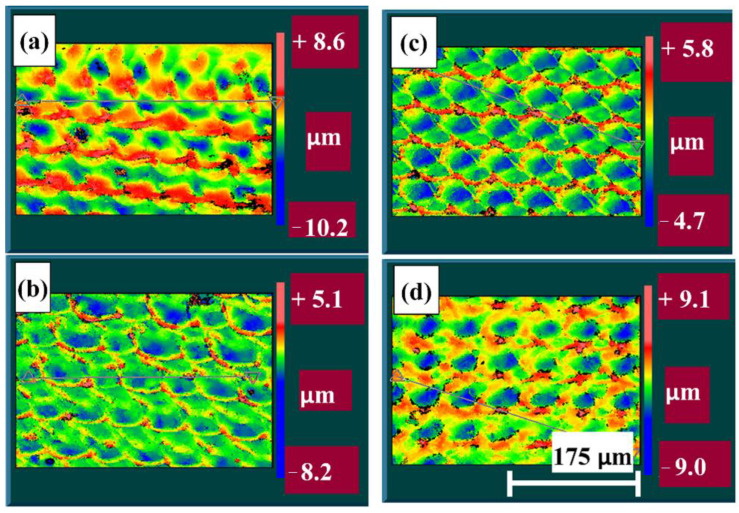
Pseudo-color image of 3D profiles from the profilometer: (**a**) AlTiN1; (**b**) AlTiN2; (**c**) AlTiN3 and (**d**) AlTiN4.

**Figure 6 materials-14-00951-f006:**
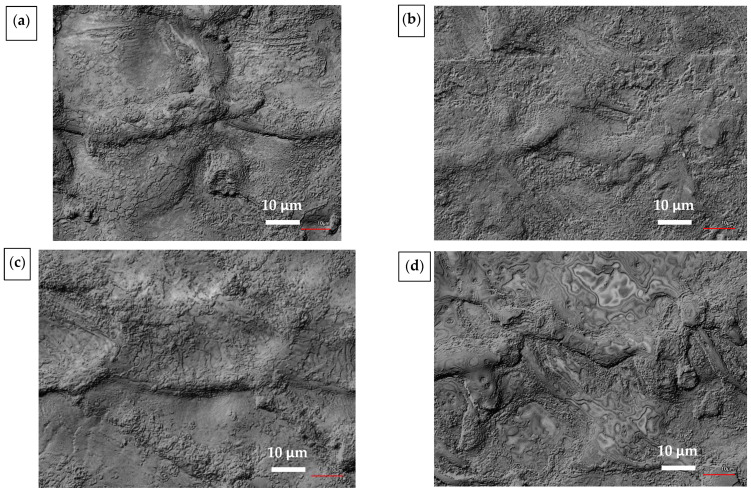
SEM images of stripped samples: (**a**) AlTiN1; (**b**) AlTiN2; (**c**) AlTiN3; (**d**) AlTiN4.

**Figure 7 materials-14-00951-f007:**
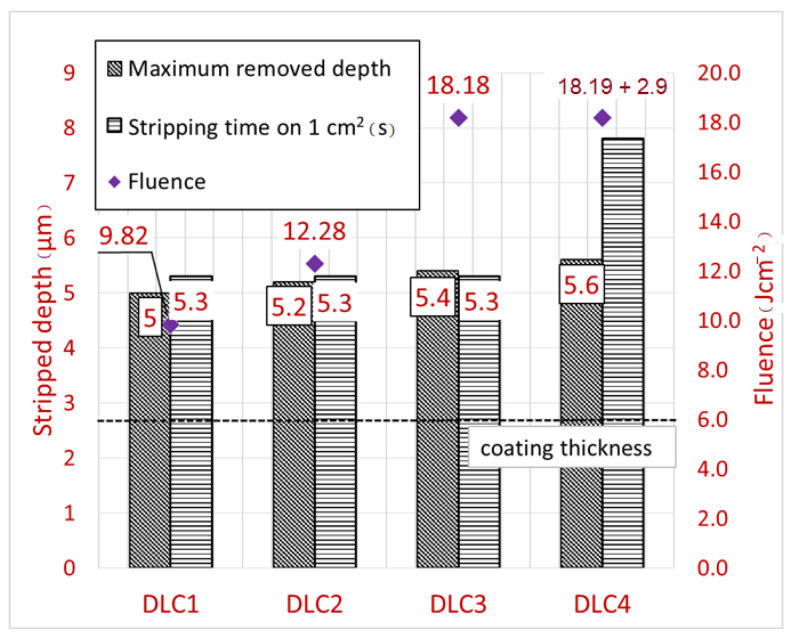
Bar chart of removed depth and working time for laser stripping of DLC C coatings.

**Figure 8 materials-14-00951-f008:**
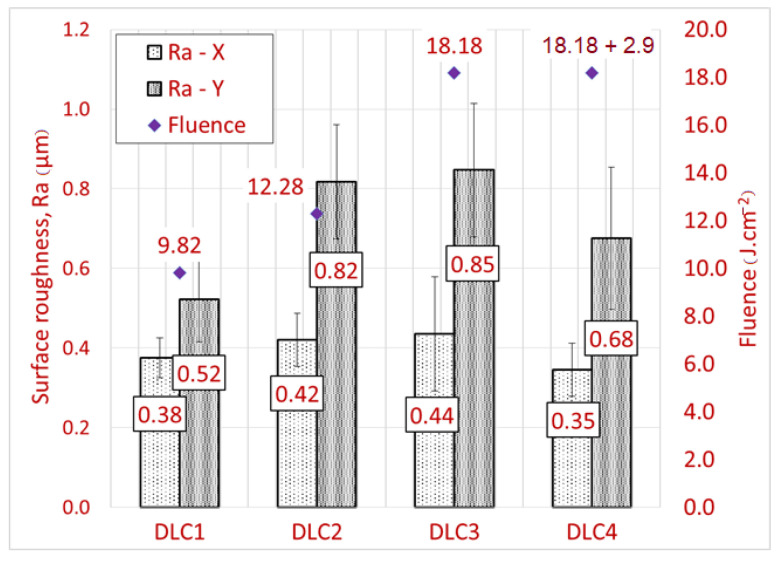
Bar chart of surface roughness after laser stripping of the DLC C coatings in relation to the fluence.

**Figure 9 materials-14-00951-f009:**
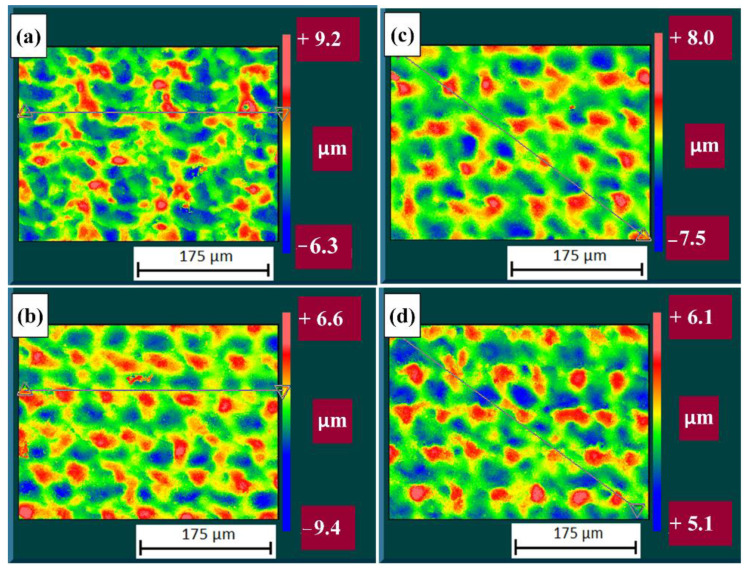
Pseudo-color image of 3D profiles from the profilometer: (**a**) DLC1; (**b**) DLC2; (**c**) DLC3 and (**d**) DLC4.

**Figure 10 materials-14-00951-f010:**
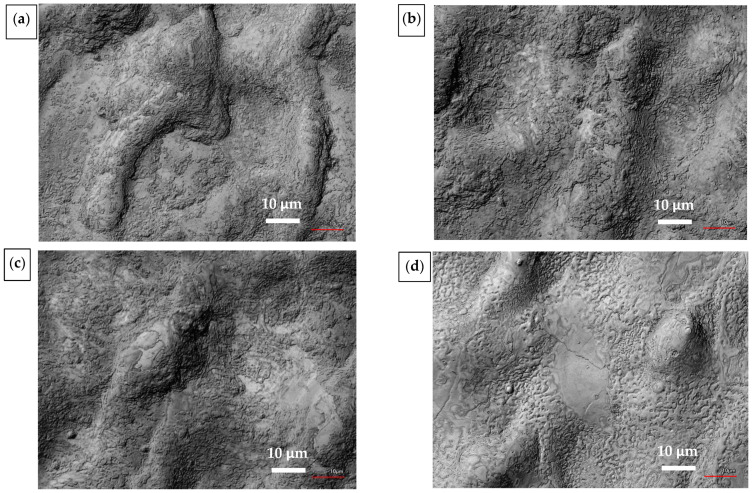
SEM images of stripped samples: (**a**) DLC1; (**b**) DLC2; (**c**) DLC3; (**d**) DLC4.

**Table 1 materials-14-00951-t001:** Main properties of coatings used in the experiment.

Coating	AlTiN	DLC C
Substrate	1.2379 (X155CrVMo12) steel	1.2379 (X155CrVMo12) steel
Coating hardness	3000–3600 HV	850–950 HV
Thickness	3.2 µm	2.6 µm
Special properties	High toughness, high thermal stability—operational temperature max. 900 °C	COF against steel 0.08; multilayer coating: CrN 1.1 µm, top layer a-C:H 1.5 µm

Both coatings were deposited by a PVD magnetron sputtering method on non-heat-treated tool steel 1.2379 (X155CrVMo12).

**Table 2 materials-14-00951-t002:** Composition of the X155CrVMo12 tool steel substrate.

Element	C	Si	Mn	Cr	Mo	V
%	1.5–1.6	0.1–0.4	0.15–0.45	11–12	0.6–0.8	0.9–1.1

**Table 3 materials-14-00951-t003:** Parameters of the laser device.

Beam Source	Nd:YAG
Wavelength (nm)	1064
Average power (P) (W)	50
Maximum pulse energy (Ep) (mJ)	40
Pulse duration (ns)	120
Spot diameter (mm)	0.18
Max. scanning speed (mm·s^−1^)	3000
Maximum repetition rate (Hz)	50,000

**Table 4 materials-14-00951-t004:** AlTiN coating laser stripping parameters.

Sample	AlTiN1	AlTiN2	AlTiN3	AlTiN4	
Frequency (Hz)	10,000	8000	5000	5000	50,000
Pulse overlap—H_x_ (%)	70	70	70	70	70
Pulse overlap—H_y_ (%)	70	70	70	70	70
Scanning speed (mm∙s^−1^)	540	432	270	270	2700
Energy in pulse (µJ)	3700	4625	7400	7400	740
Fluence (J∙cm^−2^)	14.54	18.18	29.08	29.08	2.91
Peak power density (MW∙cm^−2^)	121.17	151.46	242.33	242.33	24.23

**Table 5 materials-14-00951-t005:** DLC C coating laser stripping parameters.

Sample	DLC1	DLC2	DLC3	DLC4	
Frequency (Hz)	8000	8000	8000	8000	50,000
Pulse overlap—H_x_ (%)	70	70	70	70	70
Pulse overlap—H_y_ (%)	70	70	70	70	70
Scanning speed (mm·s^−1^)	432	432	432	432	2700
Energy in pulse (µJ)	2500	3125	4625	4625	740
Fluence (J·cm^−2^)	9.82	12.28	18.18	18.18	2.91
Peak power density (MW·cm^−2^)	81.869	102.34	151.46	151.46	24.23

**Table 6 materials-14-00951-t006:** Area surface roughness parameters after stripping of AlTiN coating.

Roughness Parameters	Sa (µm)	Sz (µm)	Ssk	Sku
AlTiN1	1.28 ± 0.14	6.28 ± 0.2	0.11	2.61
AlTiN2	0.52 ± 0.04	3.66 ± 0.46	0.45	3.17
AlTiN3	1.29 ± 0.12	6.61 ± 0.29	0.56	2.94
AlTiN4	1.18 ± 0.1	6.66 ± 1.16	0.59	3.60

**Table 7 materials-14-00951-t007:** Amounts of elements on the surface after laser stripping of the AlTiN coating.

Part of a Sample	Element (Wt%)	AlTiN1	AlTiN2	AlTiN3	AlTiN4
Coating	N	3.62 ± 0.31	1.16 ± 0.16	2.43 ± 0.20	1.01 ± 0.17
	Ti	10.1 ± 0.24	12.81 ± 0.22	16.27 ± 0.36	18.14 ± 0.33
	Al	8.68 ± 0.17	9.53 ± 0.15	13.23 ± 0.28	13.69 ± 0.24
Substrate	V	0.61 ± 0.13	0	0	0
	O	19.57 ± 0.40	14.45 ± 0.23	29.61 ± 0.61	26.84 ± 0.46
	Si	0.64 ± 0.04	0.37 ± 0.03	0.55 ± 0.04	0.40 ± 0.03
	Cr	12.59 ± 1.52	14.6 ± 1.13	12.58 ± 1.71	12.14 ± 1.42
	Fe	44.2 ± 0.84	47.08 ± 0.67	25.34 ± 0.61	27.78 ± 0.53

**Table 8 materials-14-00951-t008:** Area surface roughness parameters after laser stripping of DLC C coating.

Roughness Parameters	Sa (µm)	Sz (µm)	Ssk	Sku
DLC1	1.47 ± 0.15	8.07 ± 0.43	0.65	3.34
DLC2	1.30 ± 0.27	6.58 ± 0.42	0.32	2.99
DLC3	1.23 ± 0.10	6.48 ± 1.08	0.54	3.38
DLC4	1.16 ± 0.11	5.83 ± 0.53	0.43	3.26

The samples were further analysed by EDS/SEM.

**Table 9 materials-14-00951-t009:** Amounts of elements on the surface after laser stripping of the DLC C coating.

Part of a Sample	Element [Wt%]	DLC1	DLC2	DLC3	DLC4
Coating	C	15.63 ± 0.26	10.7 ± 0.16	12.07 ± 0.17	3.19 ± 0.07
	N	0.69 ± 0.13	0.65 ± 0.09	0.59 ± 0.10	-
	Cr	31.16 ± 0.79	29.87 ± 0.63	28.37 ± 0.65	40.66 ± 0.66
Substrate	O	9.33 ± 0.16	7.57 ± 0.11	8.04 ± 0.12	13.73 ± 0.19
	V	0.36 ± 0.08	0.37 ± 0.07	0.41 ± 0.07	0.44 ± 0.07
	Mo	0.36 ± 0.08	0.48 ± 0.07	0.49 ± 0.07	0.22 ± 0.07
	Si	0.21 ± 0.02	0.20 ± 0.02	0.21 ± 0.02	0.17 ± 0.02
	Mn	3.04 ± 0.92	3.19 ± 0.76	3.35 ± 0.77	8.11 ± 0.76
	Fe	39.21 ± 0.63	46.98 ± 0.58	46.47 ± 0.59	33.48 ± 0.50

## Data Availability

Data is contained within the article.

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
