# Peer review of "Investigation of Multiparameter Laser Stripping of AlTiN and DLC C Coatings"

_materials, 2021, doi:10.3390/ma14040951_

Round 1

Reviewer 1 Report

The paper presents results obtained using a method for coating removal by laser stripping. Two different hard coatings used in cutting tools or forming mould surfaces are studied. The authors state that the novelty of the article is in the achievement of short processing times resulting in a productive method. Various parameters such as repetition rate, scanning speeds and energies per laser pulse are modified, and the results are analyzed using optical microscopy and EDS/SEM. The laser has a relatively long pulse duration, and IR radiation is used; both of these characteristics lead to larger thermal effects in laser microprocessing, which are detrimental.

Although the “stripping” times given in the paper are indeed short,  chemical characterization of the samples indicate that about 30% of the coating material is still present after the irradiation, even though thicknesses greater than the thickness of the coating are ablated. In fact, the authors themselves state that “The results of the analysis show that there are coating residues on the stripped surface. There are islands of coating, residual oxide created by melting and other impurities on the decoated areas.” The coating is therefore not completely removed; only 70% of the coating is removed, as estimated by the authors. In addition, the chemical composition of the irradiated surfaces are modified, as stated by the authors themselves: “However, all of the surfaces contained a significant percentage of oxygen, in the form of oxides with the elements mentioned above. The oxides are most likely waste generated during the laser stripping process.” In these  conditions, comparisons with other authors which have obtained longer processing times are not significant, since the authors cited have obtained complete stripping of the coating without affecting the underlying material, which the authors of the paper under review have not. Therefore, the paper should be considered for publication only if the authors can prove that the surface can successfully be recoated after the laser stripping they present. Otherwise, the short processing times are irrelevant. These short processing times are the only novelty of the paper.

Other problems are given below.

  • Figure 1 seems unnecessary. It contains no information in addition to that given in the text.
  • What do the authors mean by “Gaussian beam effect” (line 141-142)? Related to this, how exactly was the 70% spot overlap chosen? The authors only state that this was chosen based on previous experience, without giving any details.
  • What is the difference between the beam diameter w0 and the spot diameter D?
  • Why does the factor 2 appear in equation 1 if w0 is a diameter (see, for example, [9], or other papers such as Applied Sciences 2019,9,3962;Materials 2020, 13, 969)?
  • The fluence is modified by modifying the power and the repetition rate. Does the modification of the power affect the spot diameter on the target?
  • Referring to figure 3, the authors state: “In summary, an increase in fluence may lead to an increase in the removed depth”. This, however, is contradicted by the results in fig. 3, where as the fluence increases, the removed depth is 4.6; 4; 4.4. As in the case of DLC removal, there doesn’t appear to be a relation between the fluence and the removed depth.
  • If the error bars in figures 4 and 7 for the roughness along x are taken into account, there is no clear dependence of the surface roughness on the fluence, either.
  • The bibliography should be checked and Among other things, some of the pages of the articles cited seem to have been given randomly (for example [9], [10]. [11], [13],………).

Several minor correction related to the text should also be made:

-there are several error messages (ref source not found) which should be addressed

- some correction of English is needed. For example in row 102, it should be disk thickness, not height

- I think that in “DLC C coatings” the second C is redundant; DLC is diamond like carbon.

Author Response

Dear reviewer,

we would like to thank you for your revision of our manuscript. We tried to do our best to find the right answers to your questions. The respons to your comments can be found bellow and also in the attached document

  • We thank reviewer for a significant observation, which can improve the quality of our manuscript. In our case, optical analysis show complete stripping. In addition, according to the coating provider samples were recoated without any difficulties. Based on this observation, processing times are relevant and we believe that our findings will encourage more research in this field to speed up laser stripping. However, we take the reviewer comment seriously and we will examine in detail the effect of coating residues detected by EDS on recoating process in our future work.

Other problems are given below.

  1. Figure 1 seems unnecessary. It contains no information in addition to that given in the text.
    • The aim of this figure is to show the appearance of the coated samples. We agree with your argument and this figure can be removed from the paper.
  2. What do the authors mean by “Gaussian beam effect” (line 141-142)? Related to this, how exactly was the 70% spot overlap chosen? The authors only state that this was chosen based on previous experience, without giving any details.
    • The spot overlap was chosen to be 70% as a compromise between the energy transferred to the material and the depth of removal. Moreover, spot overlap of 70% allow us to use higher scanning speed. According to the results of this article, we will study in depth the overlap of spots for further publications.
  3. What is the difference between the beam diameter w0and the spot diameter D? Why does the factor 2 appear in equation 1 if w0 is a diameter (see, for example, [9], or other papers such as Applied Sciences 2019,9,3962;Materials 2020, 13, 969)?
    • An error has occurred here. The parameter w0 was to mean the radius of the beam. Only parameter D remained in the article.
  4. The fluence is modified by modifying the power and the repetition rate. Does the modification of the power affect the spot diameter on the target?
    • Yes, it can, but the energies used are relatively high, so the effect of energy on the spot diameter is low.
  5. Referring to figure 3, the authors state: “In summary, an increase in fluence may lead to an increase in the removed depth”. This, however, is contradicted by the results in fig. 3, where as the fluence increases, the removed depth is 4.6; 4; 4.4. As in the case of DLC removal, there doesn’t appear to be a relation between the fluence and the removed depth.
    • The relation between fluence and removed depth for both coating were better explained.
  6. If the error bars in figures 4 and 7 for the roughness along x are taken into account, there is no clear dependence of the surface roughness on the fluence, either.
    • A trend of the surface roughness was confirmed by the surface area roughness measurement and this dependence was better explained in discussion.
  1. The bibliography should be checked and Among other things, some of the pages of the articles cited seem to have been given randomly (for example [9], [10]. [11], [13],………).
    • The bibliography was renumbered and checked.
  2. Several minor correction related to the text should also be made:
  3. -there are several error messages (ref source not found) which should be addressed
    • The content has been updated and missing references have been added.
  4. some correction of English is needed. For example in row 102, it should be disk thickness, not height
    • The text was checked again by native speaker.
  5. I think that in “DLC C coatings” the second C is redundant; DLC is diamond like carbon.
    • Yes, you're right, the second C is redundant, but our coating provider uses this indication for clarity to know that there are no other additives in the coating. For example, it produces a coating called DLC W, which contains tungsten, etc.
    •  

Reviewer 2 Report

Reviewer’s Comments on the 1097433 Manuscript

            In this study, the authors propose a method of multiparameter laser stripping of AlTiN and DLC C coatings from the surface of tool steel substrate. Authors report a good surface quality after the rapid laser stripping and support their claim with materials characterization techniques such as EDS, SEM, and 3D profilometry. Although the manuscript is already in a good shape, there are some minor points that have to be addressed before making the final decision.

  1. [Page 2, Line 92; Page 3, Lines 97, 102, 110; Page 4, Lines 125, 135] Please add the missing references.
  2. [Page 3, Line 106] Is there a specific reason, why authors have chosen 1064 nm wavelength laser? Or is it just the device’s default setting?
  3. As a general comment, authors should use superscripts for the units in the legends of their figures 3 and 5.
  4. EDS is a strong technique for rapid chemical composition analyses, however it is known to perform with a high error margin for elements having atomic number smaller than Na which is “11”. Under the light of this knowledge, reviewer would like authors to conduct chemical composition analyses of these samples with and another analytical technique such as wavelength dispersive X-Ray spectroscopy (WDS), optical emission spectroscopy (OEM), or x-ray photoelectron spectroscopy (XPS).

Overall, the reviewer found this study of important as it introduces a method for rapid and effective removal of surface coatings from substrates with negligible damage. The authors have done a good job in terms of the explanation of the experimental details and evaluation of the test results. The reviewer only found a few minor issues and a major issue that need to be addressed, then I believe the manuscript will be ready for publication and will capture more attention in the community.

Author Response

Dear reviewer,

we would like to thank you for your revision of our manuscript. We tried to do our best to find the right answers to your questions. The respons to your comments can be found bellow and also in the attached document

  1. [Page 2, Line 92; Page 3, Lines 97, 102, 110; Page 4, Lines 125, 135] Please add the missing references.
    • The content has been updated and missing references have been added.
  2. [Page 3, Line 106] Is there a specific reason, why authors have chosen 1064 nm wavelength laser? Or is it just the device’s default setting?
    • There is no specific reason to use this laser wavelength. This wavelength is the default for this system. For more experiments on this topic, we plan to use a new laser system with possible generation of second (515 nm) and third harmonic (343) wavelengths to obtain different absorption and interaction processes, but this system works with femtosecond pulses, so ablation mechanisms and contribution will be other.
  3. As a general comment, authors should use superscripts for the units in the legends of their figures 3 and 5.
    • The superscripts have been added to these figures.
  4. EDS is a strong technique for rapid chemical composition analyses, however it is known to perform with a high error margin for elements having atomic number smaller than Na which is “11”. Under the light of this knowledge, reviewer would like authors to conduct chemical composition analyses of these samples with and another analytical technique such as wavelength dispersive X-Ray spectroscopy (WDS), optical emission spectroscopy (OEM), or x-ray photoelectron spectroscopy (XPS).
    • We thank reviewer for a significant observation, which can improve the quality of our manuscript. The mentioned articles [9, 10, 11] analysed surface after stripping with SEM/EDS methods only. In addition, according to the coating provider samples were recoated without any difficulties Based on this observation, processing times are relevant and we believe that our findings will encourage more research in this field to speed up laser stripping. In addition, we take the reviewer comment seriously and we will examine in detail the effect of coating residues detected by other chemical analysis in our future work.

Reviewer 3 Report

Dear Authors,

after reading through the paper proposed for publication in Materials, some minor and major comments arose. Here you can find some minor comments

Line 85 – 86:

the mechanical properties of AlTiN, as for the majority of PVD coatings, are highly dependent on processing parameters (e.g. N2 flow and pressure, bias voltage…) that influence both the coating structure as its micro/nano structure. I consider it would be more appropriate to indicate an interval of hardnesses ranging approx.. between 2200 and 3000 HV.

Line 92:

Typo error. Please correct the missing reference (reference not found), such errors are present along the length of the whole paper, please correct.

Table 1

Were the mechanical properties of the coating measured in some way? In case such values have been reported from a technical datasheet, it should be pointed.

Fig. 5 and Fig. 8 have very low quality, for example it is not possible to read numbers in the scales on the right side.

Generally speaking I consider the English language adopted in the paper, adequate for a scientific publication, nevertheless some major modification are strongly suggested to make the paper more solid.

The Discussion chapter would be more appropriate for a Conclusions chapter. Try to restructure that part making some comparison with similar studies from literature, as you point out in the introduciton, similar studies were already made in literature. Try to compare for examples the degree of surface finishing you obtained.

Something I was seraching for, in your paper, is an evaluation of the degree of surface integrity after laser stripping. You wrote you've made SEM characterization so it would be important to add those characterization to the results obtained by EDS in order to strengthen that discussion. The images with the optical profilometer are not enough to give evidence of the morpology  of the surface.

Finally, due to the fact that you are analysing a 3D surface it would be more appropriate and would carry more information, the use of 3D surface topology indicators of the 3rd and 4th order such as Ssk and Sku. Consider going more in depth with this kind of approach because it can be more meaningful for the kind of surfaces you include in this analysis.

Kind regards

Author Response

Dear reviewer,

we would like to thank you for your revision of our manuscript. We tried our best to find suitable answers to your questions. The respons to your comments can be found bellow and also in the attached document

  1. Line 85 – 86:

the mechanical properties of AlTiN, as for the majority of PVD coatings, are highly dependent on processing parameters (e.g. N2 flow and pressure, bias voltage…) that influence both the coating structure as its micro/nano structure. I consider it would be more appropriate to indicate an interval of hardnesses ranging approx.. between 2200 and 3000 HV.

  • Yes, you are right. It is better to indicate an interval of hardness. These intervals were added into the table according our coating provider. Moreover I am adding the website of the coating provider. https://www.advamat.cz/dlc-c/
  1. Line 92:

Typo error. Please correct the missing reference (reference not found), such errors are present along the length of the whole paper, please correct.

  • Missing references have been added.
  1. Table 1

Were the mechanical properties of the coating measured in some way? In case such values have been reported from a technical datasheet, it should be pointed.

  • Mechanical properties were taking from a technical datasheet. The depth of the coating was used in conjunction with the scratch test to evaluate the thickness and cohesion of the coating and substrate. This information has been added to the article.

  1. 5 and Fig. 8 have very low quality, for example it is not possible to read numbers in the scales on the right side.
    • Images were re-uploaded with higher quality. To maintain the same resolution, 3D images were removed and pseudo-color images were enlarged.
  2. Generally speaking I consider the English language adopted in the paper, adequate for a scientific publication, nevertheless some major modification are strongly suggested to make the paper more solid.
    • The text was checked again by native speaker.
    •  
  3. The Discussion chapter would be more appropriate for a Conclusions chapter. Try to restructure that part making some comparison with similar studies from literature, as you point out in the introduciton, similar studies were already made in literature. Try to compare for examples the degree of surface finishing you obtained.

  • The studies from the introduction were used in discussion chapter. Moreover some references were added into introduction and discussion too.
  •  
  1. Something I was seraching for, in your paper, is an evaluation of the degree of surface integrity after laser stripping. You wrote you've made SEM characterization so it would be important to add those characterization to the results obtained by EDS in order to strengthen that discussion. The images with the optical profilometer are not enough to give evidence of the morpology  of the surface.
  • SEM images have been added to the article and also a brief description of these pictures. Cracks were found on one of the samples. Therefore, detailed measurements of these bursts were made.
  1. Finally, due to the fact that you are analysing a 3D surface it would be more appropriate and would carry more information, the use of 3D surface topology indicators of the 3rd and 4th order such as Ssk and Sku. Consider going more in depth with this kind of approach because it can be more meaningful for the kind of surfaces you include in this analysis.
    • A 3D surface topography indicators followed by comments has been added into article.

Reviewer 4 Report

The presented work describes experimental work on laser stripping of AlTiN and DLC C coatings from 1.2379 tool steel substrate. The manuscript is well organized and results are properly presented. However, the manuscript is similar more to a report rather than scientific paper. The discussion has to be expanded and some parts described much better.

The following comments need to be addressed by the authors prior to acceptance:

  1. Introduction is well written but a number of references is rather low. It should be extended.
  2. Page 1 (last paragraph) - there is a mistake in reference numbering. Reference [6] appears prior to [3]. Please verify carefully references listed in the manuscript.
  3. Page 2 (last line before the Table 1) - there is an error message linking Table 1: Error! Reference source not found.
  4. In the sentence with mistaken reference [6] it should be added abbreviation to diamond-like coatings – DLC.
  5. Page 3-4: Again referring Table 2, 3, 4 and 5, as well as Figure 1 appears: Error! Reference source not found.
  6. In Table 2, Fe element is missing.
  7. Why did you vary the conditions of laser stripping parameters for AlTiN and DLC C, except AlTiN2 and DLC3)? It is not clear.
  8. What can be the risk of using a second low energy pass? It is mentioned that the melted material solidified again into a form with smoother surface. It is not confirmed in AlTiN4 and DLC4. Moreover, the melted material of X155CrVMo12 may solidify into steel with different compositions and in consequence with distinct mechanical characteristics. It should be discussed in more details.
  9. The total surface roughness from certain measurement area should be added as well to the X and Y results. What is the surface roughness of bare X155CrVMo12?
  10. The 3D images (surface maps) in Figure 5 should be rescaled uniformly for all samples (Z axis) for better comparison. What is the origin of a peak-valley type morphology? Is it associated with the Gaussian beam effect?
  11. What was the EDS analysis area? Please include standard deviations to Table 1 data (page 8). It should be actually Table 6. Importantly, analysing chemical composition by EDS in such rough samples might be misleading. Please clarify.
  12. Page 11 (line 313): DLC5? It should DLC4.
  13. There is missing deeper discussion (pages 11-12) with comparison to the most appropriate literature.

Moreover, English grammar has to be once again carefully revisited because there are some mistakes. It is suggested to perform comprehensive scientific proofreading and editing by native English speakers.

Author Response

Dear reviewer,

we would like to thank you for your revision of our manuscript. We tried to do our best to find the right answers to your questions. The respons to your comments can be found bellow and also in the attached document

  1. Introduction is well written but a number of references is rather low. It should be extended.
  • Some relevant references has been added into introduction and discussion too.
    •  
  1. Page 1 (last paragraph) - there is a mistake in reference numbering. Reference [6] appears prior to [3]. Please verify carefully references listed in the manuscript.
    • The bibliography was renumbered and checked.

  1. Page 2 (last line before the Table 1) - there is an error message linking Table 1: Error! Reference source not found.
    • Missing references have been added.

  1. In the sentence with mistaken reference [6] it should be added abbreviation to diamond-like coatings – DLC.
    • Missing abbreviations have been added
  2. Page 3-4: Again referring Table 2, 3, 4 and 5, as well as Figure 1 appears: Error! Reference source not found.
    • Missing references have been added.
  3. In Table 2, Fe element is missing.
    • An amount of elements in tool steel 1.2379 was taken from datasheet. The amount of Fe is given by the remainder from 100 percent from the sum of all alloying elements according to their content in the steel.
  4. Why did you vary the conditions of laser stripping parameters for AlTiN and DLC C, except AlTiN2 and DLC3)? It is not clear.
    • Both coatings are completely different, so we chose different parameters for each, with only one parameter intersecting
  5. What can be the risk of using a second low energy pass? It is mentioned that the melted material solidified again into a form with smoother surface. It is not confirmed in AlTiN4 and DLC4. Moreover, the melted material of X155CrVMo12 may solidify into steel with different compositions and in consequence with distinct mechanical characteristics. It should be discussed in more details.
    • Some changes in the underlying steel could occur, and we tried to better describe these phenomena in the discussion. Moreover, it is a good point and it will be a subject of our further studies.
  6. The total surface roughness from certain measurement area should be added as well to the X and Y results. What is the surface roughness of bare X155CrVMo12?
    • The initial disc surface roughness has been added into paper. Also area surface roughness has been added.
  7. The 3D images (surface maps) in Figure 5 should be rescaled uniformly for all samples (Z axis) for better comparison. What is the origin of a peak-valley type morphology? Is it associated with the Gaussian beam effect?
  • Images were re-uploaded with higher quality. To maintain the same resolution, 3D images were removed and pseudo-color images were enlarged. Moreover the parameters of an area surface roughness has been added.
  1. What was the EDS analysis area? Please include standard deviations to Table 1 data (page 8). It should be actually Table 6. Importantly, analysing chemical composition by EDS in such rough samples might be misleading. Please clarify.
    • The standard deviation has been added to the table. The EDS analysis area was set to 200 to 200 µm. This information has been added into the text.
  2. Page 11 (line 313): DLC5? It should DLC4.
    • This mistake has been fixed. It really was supposed to be DLC 4.
  3. There is missing deeper discussion (pages 11-12) with comparison to the most appropriate literature.
    • Several other points were added to the discussion, also taking into account the literature from the introduction. Some new studies have also been added. It should also be noted that there are not so many publications in the topic of laser stripping and it is not a fully described topic.

Moreover, English grammar has to be once again carefully revisited because there are some mistakes. It is suggested to perform comprehensive scientific proofreading and editing by native English speakers.

  • The text was checked again by native speaker.

Round 2

Reviewer 1 Report

The problems I have raised in the review have mostly been corrected. The error messages I referred to, however, have not been corrected. Also, figure 1, which the authors agreed can be removed, has not been removed.

The authors have also stated in their answer that the coating provider stated that the samples were recoated without any difficulties. I have not found this mentioned in the paper;  I think it is important for the authors to mention this in the paper, since it is very important in practice.

There are still some other text errors in the paper. For example, the bibliographical references are not given correctly in the text, and on page 8 some lines are incomplete. The English should also be checked again.

Author Response

Dear reviewer,

Thank you for the second round of revisions. You can find all changes in the revised document in MS Word in change mode. The pdf is a clean version of the document.

The respons to your suggestions are listed bellow. 

1. Figure 1 has been removed from the article. Figure numbers have been renumbered. 

2. Yes, it is very important and should be mentioned. Following your recommendation, we have added this point to the discussion.

3. The errors you mentioned on page 8 are probably problems caused by formatting the document. We think that during the preparation of the manuscript for publication, these errors will be corrected by us and the editors.

Thank you for your time in our article and for your revisions, which have increased the quality of our manuscript.

As you suggested, we had our manuscript checked by a native speaker. We now hope that our manuscript is of adequate quality for publication in the Materials journal.

Kind regards

Tomas Primus

Reviewer 2 Report

            Authors did a good job addressing most of the issues brought to attention by the Reviewer. However, the missing references is still an issue, actually except for introduction section references are not visible for the rest of the manuscript. After fixing that, paper will be ready for publication.

Author Response

Dear reviewer,

Thank you for the second round of revision.
We have tried to add as much references as we found, both to the text and to the discussion.
We think we have found and described all known and relevant literature for our article.
In total, the article contains 33 relevant sources related to the topics of laser stripping, coatings, laser ablation and more.
We now hope that the manuscript is in sufficient form for publication

In addition, the manuscript was checked by a native speaker, so we hope the quality of the manuscript has increased.

You can find all changes in the revised document in MS Word in revision mode. The pdf is a clean version of the document.

Kind regards

Tomas Primus

Reviewer 3 Report

Dear Authors,

thank You for providing the required correction. I consider the paper to be significantly improved since the first review, therofore I recommend it for publication in this Journal, after a minor check of the English language

Kind regards

Author Response

Dear reviewer,

Thank you for your time in our article and for your revisions, which have increased the quality of our manuscript.

As you suggested, we had our manuscript checked by a native speaker. We now hope that our manuscript is of adequate quality for publication in the Materials journal.

You can find all changes in the revised document in MS Word in change mode. The pdf is a clean version of the document.

Kind regards

Tomas Primus